# Vitiligo: Pathogenesis and New and Emerging Treatments

**DOI:** 10.3390/ijms242417306

**Published:** 2023-12-09

**Authors:** Javier Perez-Bootello, Ruth Cova-Martin, Jorge Naharro-Rodriguez, Gonzalo Segurado-Miravalles

**Affiliations:** Ramon y Cajal University Hospital, Road M-607, 9, 100, 28034 Madrid, Spain; jpbootello@gmail.com (J.P.-B.); ruth.cova.97@gmail.com (R.C.-M.); jorgenrmed@gmail.com (J.N.-R.)

**Keywords:** vitiligo, JAK inhibitor, pathogenesis, therapeutics

## Abstract

Vitiligo is a complex disease with a multifactorial nature and a high impact on the quality of life of patients. Although there are multiple therapeutic alternatives, there is currently no fully effective treatment for this disease. In the current era, multiple drugs are being developed for the treatment of autoimmune diseases. This review assesses the available evidence on the pathogenesis of vitiligo, and a comprehensive review of treatments available for vitiligo now and in the near future is provided. This qualitative analysis spans 116 articles. We reviewed the mechanism of action, efficacy and safety data of phototherapy, afamelanotide, cyclosporine, phosphodiesterase 4 inhibitors, trichloroacetic acid, basic fibroblast growth factor, tumor necrosis factor (TNF) inhibitors, secukinumab, pseudocatalase and janus kinase (JAK) inhibitors. At the moment, there is no clearly outstanding option or fully satisfactory treatment for vitiligo, so it is necessary to keep up the development of new drugs as well as the publication of long-term effectiveness and safety data for existing treatments.

## 1. Introduction

Vitiligo is a disease characterized by the appearance of white depigmented patches on the skin due to a selective loss of melanocytes [1]. It affects 0.1–2% of the world’s population, with no significant differences in gender, ethnicity or geographic region [2,3]. Although it is not a disease that shortens life expectancy, it causes a significant negative psychological impact, comparable to other skin diseases such as eczema or psoriasis [4,5].

Vitiligo is a disease that is currently classified as an autoimmune disease [6]. It is a complex disease, in which genetics plays an important, but yet not fully elucidated role [7]. Heritability—the fraction of disease risk attributable to genetic variation—is high, estimated to be in European population between 0.75 and 0.83 [8]. More than 50 susceptibility loci for vitiligo have been discovered [7]. However, the occurrence of vitiligo is not solely explained by genetic factors: instead, the convergence theory proposes that vitiligo occurs as a result of the interaction between immunological, biochemical and environmental factors in genetically predisposed patients [9].

There are several reviews of the literature on medical treatments for vitiligo. These include reviews of specific emerging drug groups such as anti-JAK drugs [10], the emergence of new drugs in animal research [11] or global considerations in vitiligo [12].

Herein, we provide an updated review of the molecular pathogenesis of vitiligo, as well as a review of new treatments currently being studied from a clinical perspective without losing the molecular approach.

## 2. Pathogenesis of Vitiligo

### 2.1. Autoimmunity

Cell-mediated immunity drives vitiligo.

It is postulated that the main mechanism by which vitiligo is initiated and perpetuated is cell-mediated immunity [13]. There is ample evidence to support this theory, such as the existence of CD8+ T-cell infiltrates in skin biopsies taken at the margin of vitiligo lesions [14]. In addition, patients with vitiligo have a higher number of CD8+ T cells that are autoreactive to melanocyte-specific antigens such as tyrosinase, Melan-A/MART-1, gp100, TRP-1 and TRP-2 [15]. These autoreactive cells have been shown to induce vitiligo-like lesions in autologous skin tissue ex vivo [16]. Finally, another finding that supports the involvement of cell-mediated immunity is the high prevalence—up to 4%—of vitiligo-like lesions in patients treated with immunotherapy for melanoma [17]. This immunotherapy blocks T-cell checkpoint inhibitors, thus allowing these T cells to act uncontrollably on melanocytes [18].

Vitiligo pathogenesis acts through the activation of the JAK-STAT pathway.

The action of these T cells is highly dependent on the interferon gamma chemokine (IFN-γ) axis [19]. T cells secrete IFN-γ, which induces the production of chemokines CXCL9 and CXCL10 by keratinocytes [20]. These chemokines bind to the T-cell receptor CXCR3, increasing T-cell recruitment, thereby leading to the initiation, progression and maintenance of vitiligo lesions [21].

The IFNγ pathway is related to the JAK-STAT pathway through IFNγ binding to a specific cell surface receptor (IFNγR), which forms a heterodimeric protein that activates JAK proteins via phosphorylation. JAK proteins phosphorylate STAT, thereby activating it. Phosphorylated STAT proteins translocate to the nucleus and act as a transcription factor, thereby binding to DNA, regulating transcription of a variety of genes and affecting cell growth and apoptosis [22]. This is the physiological basis of JAK inhibitors’ utility in the treatment of vitiligo [23].

Resident memory T cells play an important role in the persistence and relapse of vitiligo.

In addition, the importance of resident memory T cells (T_RM_ cells) has been identified in multiple studies. These are a type of CD8+ T cells expressing CD69, CD103 and CD49a that have the ability to remain in tissues and induce early immune responses [24]. It has been shown that there is a large population of T_RM_ cells in vitiligo skin, which appear to be involved in relapse induction by recruiting circulating T cells through the release of IFN-γ pathway cytokines [25].

IL-15 has a central role in the maintenance and function of T_RM_ cells.

IL-15 and IL-7 are key cytokines in the generation and maintenance of memory T cells [26]. IL-15-deficient mice show reduced production of T_RM_ cells, and IL-15 promotes TRM cell function ex vivo.

Therefore, blockade of one of the IL-15 receptor subunits (CD122) [27] has been tested in murine models of vitiligo with a specific monoclonal antibody, leading to a decrease in short-term effector function of T_RM_ cells through a reduction in IFNγ production, resulting in significant repigmentation in the group of mice treated. In addition, long-term depletion of T_RM_ cells, as well as other memory T-cell pools, was observed [21]. This is a mechanism that has been explored in the search for new therapeutic targets.

Humoral immunity does not play a central role in the pathophysiology of vitiligo.

On the other hand, humoral immunity does not appear to be fundamental for the pathogenesis of vitiligo, as serum titers of melanocyte-reactive antibodies do not correlate with disease activity [28] and the uniform distribution of circulating antibodies cannot explain the patchy appearance of vitiligo lesions [29].

### 2.2. Oxidative Stress

Another key element in the pathogenesis of vitiligo is the impact of oxidative stress. Available evidence suggests that oxidative stress may be the initial event that triggers the appearance of lesions [30]. Vitiligo patients’ melanocytes are more sensitive to oxidative stress, and they are more difficult to culture than melanocytes from healthy controls [31].

The skin of patients with vitiligo shows alterations in the antioxidant system.

Melanocytes release reactive oxygen species (ROS) in response to cellular stress. In addition, melanogenesis itself is an active process that generates a pro-oxidant state in the skin [32]. All this leads to an imbalance between pro-oxidants and anti-oxidants. Pro-oxidant molecules and enzymes are favored, such as superoxide dismutase (responsible for the degradation of the O_2_^−^ radical into H_2_O_2_ and O_2_) and xanthine oxidase, whereas there is a total or functional deficit of antioxidants, such as catalase (which transforms H_2_O_2_ into H_2_O and O_2_) [33].

This increases cells’ susceptibility to external and internal oxidants, leading to structural damage to DNA, proteins and lipids, as well as cell organelles such as mitochondria [34], with a functional impairment of the melanocyte.

ROS production triggers the activation of protective molecular pathways.

Oxidative stress to which melanocytes are exposed alters the protein folding machinery of the endoplasmic reticulum, leading to the accumulation of defective peptides and activating a cellular stress phenomenon known as the “unfolded protein response” (UPR). ROS also trigger the overexpression of calcium-channel-related proteins such as CGRP (calcitonin gene-related peptide) and TRPM2 (transient receptor potential cation channel subfamily M member 2), which are involved in mitochondria-dependent melanocyte apoptosis [33].

Melanocytes in vitiligo have also been shown to be deficient in protective pathways against oxidative stress such as the nuclear factor E2-related factor (Nrf2)-p62 pathway, which makes them more vulnerable to the presence of ROS [1].

Oxidative stress leads to T-cell activation.

In connection with the other fundamental pathogenic mechanism already mentioned, oxidative stress in patients with vitiligo leads to an increase in local levels of the cytokine CXCL16 due to the activation of the UPR in stressed keratinocytes. This increased expression of CXCL16 leads to the recruitment of CD8+ CXCR6+ T cells, whose expression is accompanied by loss of melanocytes in vitiligo patients [35]. A similar T-cell recruitment phenomenon occurs upon release of CXCL12 and CCL5 from melanocytes under oxidative stress, as demonstrated in animal models [36].

Finally, the presence of oxidative stress leads to a decrease in WNT expression, which negatively affects melanocyte differentiation, especially in skin affected by vitiligo in ex vivo skin models [37].

Oxidative stress may be responsible for the presence of the Koebner phenomenon in vitiligo.

Oxidative damage may be the mechanism that explains the Koebner phenomenon in vitiligo. According to this model, chronic mechanical stimulation of susceptible skin leads to increased release of oxidative particles. This oxidative damage alters the expression of cadherins that bind keratinocytes and melanocytes, thereby decreasing melanocyte adhesion and inducing the appearance of depigmented lesions [38].

## 3. Literature Search

For the first part of the narrative review (introduction and pathogenesis of vitiligo), a PubMed search was performed using the terms “Vitiligo”, “Vitiligo AND pathogenesis” and “Vitiligo AND genetics”. A collaborative selection was carried out among all authors of the most relevant articles related to this topic. Original articles, systematic and narrative reviews, guidelines and protocols were included. All sources with a similar level of evidence were analyzed, compiled and structured, and they are summarized in Section 1 of this review.

An expert consensus on vitiligo has recently been published, which provides a brief review of the therapeutic tools available for the treatment of vitiligo and makes specific recommendations for its use [39]. This document provides recommendations for the use of several well-established therapeutic modalities in the treatment of vitiligo: topical corticosteroids, topical immunomodulators, phototherapy, home light therapy, oral steroid minipulses, surgery and depigmenting therapies. These therapies are already established in routine clinical practice, so for the design of this review, their individual discussion was not considered. Despite this, the terms related to these therapies were included in the literature search so as not to rule out articles discussing the combination of these traditional modalities with other newer therapies. These conventional treatments will be briefly discussed at the end of this review.

For the literature search, the drugs mentioned in this expert consensus as drugs with little available evidence (methotrexate, cyclosporine, JAK inhibitors, anti-TNF α, IL-17 inhibitors and catalase) were included, as well as drugs mentioned in other reviews on the treatment of vitiligo [11] and other treatments known to us from our own clinical experience or from having been presented at medical congresses related to the subject.

Therefore, for the second part of the narrative review (new and emerging treatments), a PubMed search was conducted using the terms: (vitiligo) AND (afamelanotide OR JAK inhibitor OR ritlecitinib OR baricitinib OR ruxolitinib OR fluorouracil OR methotrexate OR FGF OR fibroblast growth factor OR laser OR apremilast OR crisaborole OR phosphodiesterase-4 inhibitor OR home light therapy OR home light phototherapy OR trichloroacetic acid OR THF inhibitor OR secukinumab) on 12 June 2023.

Firstly, the titles and abstracts of the articles obtained in the first search were reviewed to assess relevant studies. The inclusion criteria were (1) studies written in English or Spanish; (2) studies addressing effectiveness, tolerability and adverse effects of approved, off-label and under-research treatments for vitiligo. Systematic and narrative reviews, guidelines, protocols and conference abstracts were excluded. Articles prior to the year 2003 were excluded. Articles reporting surgical and non-surgical procedures except phototherapy were also excluded.

Secondly, the full text of articles that met the inclusion criteria was reviewed. Previous systematic and narrative reviews were examined to ensure the accuracy of our search and to manually check their reference lists. Figure 1 shows the article selection process used for this narrative review.

## 4. Conventional Treatments

### 4.1. Phototherapy

Phototherapy in its different modalities (narrow-band ultraviolet B (NB-UVB), photochemotherapy, home-based and excimer or laser devices) has been a mainstay in the treatment of vitiligo for decades [40]. Its effect seems to be justified by the induction of T-lymphocyte apoptosis, the down-regulation of inflammatory cytokines (decreasing CXLCL9 and CXCL10 expression at keratinocytes) and the up-regulation of interleukin-10, which induces T-regulatory lymphocyte differentiation. Phototherapy also decreases the number of intraepithelial Langerhans cells and induces tyrosinase activity, thereby increasing melanin production as well as melanocyte proliferation and migration from epidermal hair follicles, favoring repigmentation of the affected area [41].

Throughout the last century, the efficacy of phototherapy in vitiligo, especially psolaren-UVA (PUVA) and NB-UVB, has been widely reported [39]. In recent years, scientific efforts have been especially oriented toward studying the efficacy of the emerging home-based phototherapy as well as targeted phototherapy.

Regarding the use of home-based NB-UVB phototherapy, a single-branched study conducted by Khandpur et al. [42] demonstrated its utility in vitiligo, while Eleftheriadou et al. [43] showed its efficacy compared to a placebo. When compared to hospital-based NB-UVB, similar efficacy has been demonstrated based on various endpoints (VASI, VitiQoL, BSA) in different clinical trials [44,45,46] and a retrospective study [47]. In one of the studies, patient-perceived satisfaction was significantly lower in home-based phototherapy [47]. Regarding intervention cost, all studies are consistent in reporting that home-based NB-UVB is cheaper in the long term despite the initial investment required, especially after 3 months of treatment [44]. Some studies showed an increase in intervention-related adverse effects (especially grade 3 erythema or burns) with home-based phototherapy [44,46].

Regarding targeted phototherapy, Raghuwanshi et al. [48] reported a 37% moderate response and a 4.5% excellent response in 134 patients with localized vitiligo treated weekly for 11 weeks with targeted NB-UVB. The use of an excimer lamp could be especially useful in localized vitiligo on the face, as recently reported by Juntongjin et al. [49]. A study involving 44 patients found no statistically significant difference in the efficacy of home-based NB-UVB when compared with a hospital excimer lamp [50].

In conclusion, phototherapy remains a cornerstone in the management of vitiligo, especially in generalized vitiligo, as confirmed by Sakhiya et al. [51] in a retrospective study including 3000 patients comparing its efficacy with that of topical treatment. Targeted phototherapy modalities appear to be a useful alternative in localized forms of vitiligo. Home-based phototherapy provides a cost-effective alternative for prolonged treatments.

### 4.2. Afamelanotide

Afamelanotide is a potent synthetic linear analogue of α-MSH in a controlled-release formulation. Subcutaneous injections result in increased skin pigmentation owing to increased expression of eumelanin. In recent years, pivotal pilot studies [52,53,54] have been undertaken to assess the effectiveness of afamelanotide in conjunction with NB–UV-B phototherapy for repigmenting non-segmental vitiligo. These studies included a comparison to NB–UV-B phototherapy as the control. During these investigations, patients underwent NB–UV-B phototherapy sessions 2–3 times weekly for 6–7 months. This regimen was either combined or not with a subcutaneously administered 16 mg implant of afamelanotide every 28 days for a duration of 4 to 6 months. The results demonstrated the superiority of the combination therapy group over the NB–UV-B monotherapy group in the experimental studies [53,54] A higher percentage of patients in the combination therapy group achieved repigmentation, and this occurred at earlier time points. In the observational study of Grimes et al. [52], a median of 66.25% repigmentation (ranging from 50% to 90%) was obtained, with 75% of cases exhibiting stability after 3 months. The most common secondary effects were hyperpigmentation along with limited cases of headaches, dizziness and nausea reported.

### 4.3. Cyclosporine

In addition to its classic role in arresting vitiligo progression, cyclosporine might be useful as an adjunctive treatment in an autologous noncultured melanocyte–keratinocyte cell transplantation (NCMKT) procedure. Although an in-depth description of NCMKT is beyond the remit of this review, the incorporation of cyclosporine has been proposed to ameliorate the depigmented halo surrounding the transplantation—an aesthetically significant sequelae frequently ascribed to the presence of CD8 T lymphocytes in perilesional skin. In accordance with this premise, Mutalik et al. [55] conducted a study involving 50 patients with stable localized vitiligo who underwent NCMKT. Of these, 25 received cyclosporine at a dosage of 3 mg/kg for 3 weeks, followed by 1.5 mg/kg for 6 weeks, in contrast to the control group, which received no adjuvant treatment. Consequently, all patients in the cyclosporine-treated group achieved a repigmentation percentage exceeding 75% (median 90.7%), whereas only seven individuals in the control group reached this threshold.

## 5. New and Emerging Therapies

### 5.1. Phosphodiesterase 4 (PDE-4) Inhibitors

PDE-4 physiologically degrades cyclic adenosine monophosphate (cAMP) to 5′-adenosine monophosphate. Inhibition of this enzyme increases intracellular cAMP and thus modifies the regulation of inflammatory mediators. These include a decrease in IL-17, 23, alfa-TNF and gamma-interferon while increasing IL-10, overall decreasing proinflammatory cytokines (which are increased in vitiligo-affected skin) and increasing suppressive cytokines [56,57]. The available evidence for PDE-4 inhibitors in vitiligo is limited to oral apremilast and a case report of topical crisaborole.

Regarding apremilast, all selected studies utilize a dose of 30 mg twice daily [57,58,59,60,61,62]. In 2019, a case series [58] and a case report [57] showed the utility of apremilast in monotherapy in vitiligo. Two clinical trials compared the improvement with the addition of apremilast to NB-UVB treatment with conflicting results [60,61]. Recently, Sharma et al. [62] have reported an improved response of vitiligo after adding apremilast to standard treatment. In most studies, headache and gastrointestinal discomfort have been reported as the main adverse effects of the treatment.

Apremilast seems to provide a comfortable option with an acceptable safety profile for vitiligo patients; however, its cost and conflicting evidence concerning its efficacy require further research in order to establish specific recommendations.

### 5.2. Trichloroacetic Acid (TCA)

The mechanism by which TCA induces repigmentation is presumably related to the ability to induce inflammation and subsequent post-inflammatory hyperpigmentation [63]. In addition, TCA-induced necrosis and subsequent trauma could theoretically stimulate melanocyte proliferation through the production of pro-opiomelanocortin and melanocortin and the release of growth factors and inflammatory mediators [64].

Regarding reported efficacy, Nofal et al. [63] published a 100-patient study reporting an 80% response rate in eyelid vitiligo, with a lower response rate in the face, torso and extremities with the application of variable concentrations of TCA every 2 weeks for 12 months. Two studies investigated the combination of microneedling and TCA, interestingly finding better response rates with the application of 70% TCA [65] than with 100% TCA [66]. The single application of 15% TCA in combination with NB-UVB achieved an excellent response in 70% of patients in a clinical trial [67]. Reported adverse effects included pain, erythema, post-inflammatory hyperpigmentation, infection and scarring.

Further research is needed on the role of TCA application in the treatment of vitiligo in monotherapy or as an adjuvant as well as to define optimal concentrations of TCA for vitiligo depending on location.

### 5.3. Basic Fibroblast Growth Factor (bFGF)

bFGF is a growth factor released by keratinocytes in response to certain stimuli. Its potential benefit in vitiligo seems to be related to its key role in melanocyte growth, migration and survival. In this regard, an increase in the release of growth factors, mainly bFGF, by keratinocytes after the application of NB-UVB has been observed [68].

Concerning its efficacy, an upgraded response has been found according to disease scales with the addition of bFGF-related decapeptide 0.1% solution treatment to tacrolimus 0.1% topical therapy in stable vitiligo [69] and to PUVA therapy [70]. Monotherapy application of bFGF-related decapeptide 0.1% solution in a retrospective study in 65 patients showed a 12% significant response at 5 months [71].

### 5.4. TNF Inhibitors

The existing data on the potential correlation between vitiligo and anti-TNF-α agents remain inconclusive due to limited evidence. Recent investigations into the utilization of anti-TNFs for vitiligo treatment have yielded controversial outcomes, characterized by a predominantly low level of evidential support in the current literature. Upon a thorough examination of the latest publications in this field, we have compiled data from case series therapeutic trials involving two patients treated with subcutaneous adalimumab [72], three patients with intravenous infliximab [72,73] and seven patients administered subcutaneous etanercept [72,74,75]. All subjects exhibited generalized vitiligo vulgaris, with the body surface area (BSA) ranging between 10% and 30%. Among these cases, there are only two reported instances wherein infliximab and etanercept demonstrated potential efficacy in vitiligo therapy. In the remaining cases, although no exacerbation in depigmented areas was observed, repigmentation was not achieved in any patient. Consequently, it can be concluded that anti-TNF-α agents have not proven effective in the treatment of vitiligo, based on current evidence. Larger-scale and long-term studies are warranted to comprehensively assess the efficacy of anti-TNF-α agents in vitiligo treatment.

### 5.5. Secukinumab

A vitiligo paradoxical adverse reaction following TNF-α agents has been documented during psoriasis treatment. The management of such cases poses a notable challenge. A recent case report describes the onset of vitiligo after the administration of adalimumab for psoriasis treatment [76], leading to its suspension and the initiation of secukinumab, an anti-interleukin-17A monoclonal antibody approved for psoriasis treatment. Complete repigmentation of vitiligo lesions and resolution of psoriasis were achieved. Despite empirical evidence affirming elevated circulating interleukin-17 (IL-17) levels and increased Th17 lymphocyte counts in vitiligo patients, along with heightened expression levels of IL-17A messenger RNA in vitiligo lesions [77], the precise role of IL-17 remains elusive. Further research is needed to elucidate IL-17’s role and ascertain its viability as a therapeutic target.

### 5.6. Pseudocatalase

The oxidative stress observed in vitiligo, related to hydrogen peroxide (H_2_O_2_)-mediated lipid peroxidation, constitutes an additional proposed mechanism contributing to the pathogenesis of vitiligo. This phenomenon has been observed in vivo through direct measurements of H_2_O_2_ levels within the depigmented epidermis. Notably, synthetic catalysts capable of oxidizing H_2_O_2_ to O_2_ and H_2_O can eliminate epidermal H_2_O_2_ [78]. One such active catalyst is a NB-UV-B-activated bis-MnII(EDTA)2(HCO_3_^−^)2 complex (EDTA, ethylenediaminetetraacetate), denoted as “pseudocatalase PC-KUS”. This topically applied pseudocatalase has been proposed as a potential agent to arrest disease progression. The most extensive patient cohort treated with this approach was documented by Schallreuter et al. Their study enrolled 71 patients presenting generalized vitiligo, categorized in two groups: control group of 10 patients subject to NB-UVB radiation and 61 individuals also receiving daily topical application of pseudocatalase. Cessation of progression was achieved in 99% of patients with low-dose PC-KUS, compared with 30% in the control group. Also, repigmentation rates above 75% were achieved in most body areas except acral areas, with results showing statistically significant differences compared with the control group.

In the most extensive patient cohort, documented by Schallreuter et al. [79], 61 individuals received daily topical application of pseudocatalase with NB-UVB radiation versus a control group of 10 patients treated with NB-UVB radiation alone. Cessation of progression was achieved in 99% of patients with low-dose PC-KUS, compared with 30% in the control group. Also, repigmentation rates above 75% were achieved in most body areas except acral areas, with results showing statistically significant differences.

These were promising results; however, subsequent clinical trials conducted by Patel et al. [80], Bakis-Petsoglou et al. [81] and Alshiyab et al. [82], among others, proved that the use of pseudocatalase was not superior to the use of placebo cream or was not associated with a therapeutic effect when combined with other treatments. In conclusion, despite initial promise, pseudocatalase has not demonstrated efficacy in the management of vitiligo, as evidenced by several clinical trials.

### 5.7. JAK Inhibitors

JAK-STAT pathway inhibition is a promising target for the treatment of vitiligo [83]. Elevated levels of interferon gamma (IFN-γ) have been observed in human skin with vitiligo, which activates via JAK 1/2 the transcription of the cytokines CXCL9 and CXCL10. These cytokines are necessary for the recruitment of cytotoxic T-lymphocytes, which are responsible for melanocyte destruction [84]. This is the reason why inhibition of JAK proteins is postulated as an effective therapeutic strategy for the treatment of vitiligo.

#### 5.7.1. Case Reports and Case Series

Clinical experience with JAK inhibitors (iJAKs) dates back to 2015 and 2016 with isolated case reports. Craiglow et al. [85] published a case report in 2015 in which an off-label therapeutic trial with oral tofacitinib (JAK 1 and JAK 3 inhibitor) was performed in a patient with generalized vitiligo, presenting almost complete repigmentation in the forehead and hands after 5 months of treatment. In 2016, Harris et al. [86] published another report in which oral ruxolitinib was initiated in a patient with coexisting vitiligo and alopecia areata. A great improvement in facial vitiligo was achieved after 20 weeks of treatment, although there was loss of pigment after withdrawal of the drug. There are other case reports in which oral tofacitinib is initiated for other pathologies and improvement of concomitant vitiligo is achieved, such as in the reports of Komnitski et al. [87] in a patient with rheumatoid arthritis and Vu et al. [88]’s report in which a marginal improvement of vitiligo was observed in a patient in whom tofacitinib was initiated for atopic dermatitis and alopecia areata. Other case reports have been published reporting improvement of vitiligo with oral iJAKs, both with tofacitinib [89] and with other drugs such as oral baricitinib, as shown in the article by Li et al. [90] in which two patients treated with baricitinib, phototherapy and topical corticosteroids and calcineurin inhibitors showed great improvement, or even upadacitinib, as in a patient with concomitant atopic dermatitis described in Pan et al.’s [91] article as having great facial repigmentation after starting this drug.

A number of case series have also been published in which oral tofacitinib was tested off-label for the treatment of vitiligo. These case series have shown heterogeneous results: Liu et al. [92] present a case series of 10 patients in which only 5 showed improvement with oral tofacitinib in sun-exposed areas or in concomitance with phototherapy. These results are consistent with those of Gianfaldoni et al. [93], who combined treatment with oral tofacitinib with phototherapy and achieved a better repigmentation rate (reaching 92% repigmentation) than with phototherapy alone. However, not all reports have yielded the same results. For instance, Fang et al. [94], in a first study, observed a poor response to treatment with oral tofacitinib and phototherapy with narrow-band UVB in four patients, although in a second study, they did find an improvement after treatment with oral tofacitinib and phototherapy with a 308 nm excimer light [95].

Finally, a series of 12 patients treated with oral upadacitinib in monotherapy has recently been published, which, in line with the results of studies with other oral iJAKs, showed moderate improvement, especially in the facial area [96].

No serious adverse effects were reported after treatment with oral iJAK, with the only adverse events reported being upper respiratory infection [92] and worsening of acne [91], with most patients being asymptomatic.

Multiple case reports and real-life case series have also been published reporting the effectiveness and safety data of topical iJAK.

In 2018, Joshipura et al. [97] reported on two patients treated with 1.5% ruxolitinib cream, who showed an improvement of vitiligo in sun-exposed areas.

Most reports, however, refer to topical iJAK tofacitinib cream in its 2% formulation. The studies by McKesey et al. [98], Mobasher et al. [99], Olamiju et al. [100] and Berbert-Ferreira et al. [101] reported the results of treatment of vitiligo with tofacitinib 2% cream in a total of 29 patients, allowing treatment with concomitant phototherapy. In all of these publications, a high degree of facial repigmentation was observed, with mixed results in extrafacial locations.

Local and systemic adverse effects have been reported with topical iJAK. Two patients treated with 1.5% ruxolitinib cream presented myalgias, which caused self-discontinuation on both of them. One of the patients even presented mild elevation of the phosphokinase level (CPK) [102]. Adverse effects with 2% tofacitinib cream were minor, such as transient erythema [101], skin contour changes on the chin [99] and acneiform lesions [99,101].

A synthesis of case reports and case series reported to date on the use of iJAK for vitiligo is shown in Table 1.

#### 5.7.2. Clinical Trials

Topical ruxolitinib, in its 1.5% cream form, is the most studied iJAK. In 2017, Rothstein et al. [103] conducted a single-group, open-label, proof-of-concept trial (NCT02809976) in which 11 patients were treated with 1.5% ruxolitinib cream. After 20 weeks, patients showed generalized improvement in vitiligo, especially the four patients with facial involvement, who showed an improvement in fVASI of 76%. In addition, an open-label extension study was performed in eight of these patients, in which the application was continued for 52 weeks and UVB phototherapy was added in three patients, improving repigmentation on the face (reaching 92%), nonacral extremities and torso, especially those treated with phototherapy [104].

In 2020, Rosmarin et al. [84] published the results of a phase 2, randomized, double-blind and dose-ranging clinical trial (NCT03099304). In this study, 157 patients were treated with ruxolitinib cream at various concentrations (from 1.5% twice daily to 0.15% once daily, plus a placebo group) for 52 weeks. Patients who achieved significance for the primary endpoint (50% improvement in facial involvement) were the patients treated with the highest concentration of ruxolitinib (1.5%). Other publications on the same clinical trial report that a 1.5% ruxolitinib cream twice daily application produces the greatest improvement in extrafacial locations (around 50% in upper and lower limbs, 15.0% in hands and 29.4% in feet) [105], and that the addition of concomitant therapy with NB-UVB phototherapy produces an additional improvement of 50.2% for F-VASI and 29.5% for T-VASI over that achieved with ruxolitinib cream monotherapy [106].

The strongest evidence for the use of ruxolitinib cream is from the TRuE-V1 (NCT04052425) and TRuE-V2 (NCT04057573) clinical trials, two multinational, phase 3, double-blind, vehicle-controlled trials of identical design involving 661 patients. The primary endpoint was to achieve F-VASI75 at week 24. In both studies, the response was clearly better in patients treated with 1.5% ruxolitinib cream twice daily. In TRuE-V1, the percentage of patients with a F-VASI75 response at week 24 was 29.8% in the ruxolitinib-cream group and 7.4% in the vehicle group (relative risk, 4.0; 95% confidence interval (CI), 1.9 to 8.4; *p* < 0.001). In TRuE-V2, the percentages were 30.9% and 11.4%, respectively (relative risk, 2.7; 95% CI, 1.5 to 4.9; *p* < 0.001). Results of key secondary endpoints showed the superiority of ruxolitinib cream over the vehicle control (F-VASI50, F-VASI90, T-VASI50 and F-VASI75 in the 52-week extension study) [107]. Evidence from the latter two clinical trials led to FDA approval of 1.5% ruxolitinib cream in 2022, applied twice daily to affected areas of up to 10% of the body surface area in adult and pediatric patients aged 12 years and older.

In all of these studies, adverse effects were mild and consisted mainly of local itching, nasopharyngitis and acne at the application site.

A clinical trial (NCT04530344) has yet to publish its final results from a 52-week extension period in 458 patients who participated in the TRuE-V1 (NCT04052425) and TRuE-V2 (NCT04057573) clinical trials to assess the long-term efficacy and safety of ruxolitinib cream in participants with vitiligo [108].

Another ongoing clinical trial (NCT05247489) is investigating the effect of the addition of phototherapy to ruxolitinib cream compared to ruxolitinib cream monotherapy [109].

In 2023, the largest clinical trial of an oral iJAK was published (NCT03715829). Treatment with ritlecitinib, a JAK 3/TEC inhibitor, was tested in a phase 2b, randomized, double-blind, placebo-controlled, parallel-group, multicenter, dose-ranging, double-blind, phase 2b study. In this study, 364 patients were randomized to once-daily oral ritlecitinib ± a 4-week loading dose (200/50 mg, 100/50 mg, 30 mg or 10 mg) or a placebo for 24 weeks (dose adjustment period). Subsequently, 187 patients received ritlecitinib at 200/50 mg daily in a 24-week extension period. Significant differences from the placebo were observed in the percentage change from baseline in the facial vitiligo area score index in the 50 mg ritlecitinib groups with (−21.2 vs. 2.1; *p* < 0.001) or without (−18.5 vs. 2.1; *p* < 0.001) a loading dose and in the 30 mg ritlecitinib group (−14.6 vs. 2.1; *p* = 0.01). Accelerated improvement was observed after treatment with 200/50 mg ritlecitinib in the extension period (n = 187). The most common adverse events were nasopharyngitis (15.9%), upper respiratory tract infection (11.5%) and headache (8.8%). Four patients had confirmed cases of herpes zoster (all non-serious), two patients had malignancies (non-melanoma skin cancers) and no thromboembolic events occurred [110]. To date, oral ritlecitinib has not yet been approved in the USA nor Europe.

In addition, a real-world clinical practice out-of-label oral tofacitinib clinical trial was conducted on 15 patients and 19 controls, in conjunction with a topical corticosteroid, topical calcineurin inhibitors and phototherapy. Both groups showed great improvement, to the extent that the differences were not statistically significant in facial lesions. No other clinical trials specific to the use of tofacitinib for vitiligo have been reported [111].

Finally, it is worth mentioning that there is an ongoing placebo-controlled dose-ranging clinical trial to evaluate the safety and efficacy of upadacitinib in subjects with non-segmental vitiligo (NCT04927975) [112].

Other trials with topical iJAK (ARQ-252 [113], cerdulatinib [114] and ATI-50002 [115]) have been initiated but no publications are available due to termination of the trial by the sponsor or lack of clear effectiveness in the preliminary results published on Clinicaltrials.gov.

A summary of the clinical trials reviewed for this section is shown in Table 2.

### 5.8. 5-Fluorouracil

5-Fluorouracil (5-FU) is a therapeutic agent that has been subject to growing interest in recent years for vitiligo treatment, thanks to the convenience of its topical formulation with very good results and almost no side effects. 5-FU stands as a crucial systemic chemotherapy agent in the treatment of cancer patients. In the context of vitiligo, its topical and intradermal formulation has been employed with diverse outcomes, often coupled with manual and electric dermabrasion, needling and fractional CO_2_ laser.

Among these approaches, microneedling has emerged as a promising technique, demonstrating favorable outcomes. This method, also known as collagen induction therapy, is a minimally invasive procedure that uses fine miniature needles to create superficial holes in the skin that are hypothesized to trigger the repair and release of growth factors, stimulate the migration of keratinocytes and facilitate the penetration of other drugs. While different therapeutic modalities have been explored, the procedural aspect remains consistent across most studies. This involves microneedling, followed by the application of a uniformly thin layer of 5% 5-FU cream or solution. This protocol is typically administered once or twice monthly for a duration ranging from 3 to 6 months. Subsequently, patients are advised to apply 5-fluorouracil cream over the same patch daily for one week following each session. Both observational [116,117,118] and experimental studies have been undertaken to compare the efficacy of combination therapy versus only microneedling [119,120,121], only 5-FU [122], and microneedling coupled with tacrolimus application [123,124] or 308 nm excimer light [125]. The overall qualitative response was better in the patches treated with the combinational therapy, 5-fluorouracil and microneedling, with statistically significant better repigmentation rates compared to those treated with tacrolimus or microneedling alone. All studies showed significantly higher and excellent responses, considered as repigmentation above 75%, and also lower (considered poor) response rates (<25%). In the study conducted by Saad et al. [125], the treatment with the combination of microneedling then application of 5-FU and excimer showed a significant and earlier response versus the excimer alone. Also, the percentage of repigmentation was higher in the patches treated with the combination, especially in the face and torso.

To facilitate the penetration of 5-FU, other techniques have also been tried, such as dermabrasion [126,127] with a similar procedure to microneedling. The most superficial layers of the skin are removed with a dermabrader until the papillary dermis is reached and then a layer of 5-FU is applied. Participants were then advised to apply topical 5% 5-FU over the abraded area once or twice daily for 2–4 weeks, with excellent repigmentation responses after the treatment. In both techniques, the most reported side effects were erythema and itching.

Recent studies have also tested intradermal infiltrations of 5-FU (50 mg/mL) every 2 weeks, comparing its effect with infiltrations of triamcinolone acetonide (3 mg/mL) with the same frequency [128]. Intradermal fluorouracil showed the best overall improvement when compared with triamcinolone. During follow-up, the vitiliginous patches continued to repigment for 6 months in fluorouracil. Finally, the combination of 5-FU with phototherapy [129] has shown better results than phototherapy alone. The main disadvantage of intradermal infiltration is the higher rate of side effects, with most patients reporting pain and a burning sensation during injections, blistering and ulcer formation.

### 5.9. Platelet-Rich Plasma

Alternative therapeutic interventions, such as platelet-rich plasma (PRP), present a regenerative treatment modality by cultivating a fertile environment rich in growth factors and cytokines. It has been proposed to stimulate the restoration of normal cellular function and holds the potential to encourage the differentiation, proliferation and maturation of melanocytes and keratinocytes, thereby contributing to epidermal repigmentation [130].

Its role has been predominantly examined in conjunction with laser [131,132], phototherapy [133] and surgical treatments [134], where it appears to exert a synergistic effect, that significantly amplifies repigmentation rates across different studies. Nevertheless, the available evidence supporting its efficacy in monotherapy remains limited. Given the current knowledge gaps, further studies are imperative to validate the effectiveness of PRP and establish comprehensive guidelines for its application in the management of vitiligo and related cases.

### 5.10. Other Regenerative Therapies

Microneedling consists of a roller with fine miniature needles used to produce micro-injuries that activate factors promoting collagen secretion by fibroblasts and stimulate melanocytes’ migration to non-pigmented areas. This technique has proven effective in monotherapy [135]. However, the combination of microneedling with topical therapies or NB-UVB was more effective compared to microneedling monotherapy [135]. As stated before, it can also be beneficial when combined with 5-fluorouracil [121]. On the other hand, microneedling has not shown an additional benefit when added to other therapies [136,137].

Other surgical therapies, oriented to grafting functional melanocytes in affected regions, constitute an emerging and promising alternative for cases of stable vitiligo. Numerous modalities have been described with different efficacy and tolerability outcomes that exceed the scope of this review and should be discussed separately [39].

### 5.11. Conventional Therapies versus Emerging Therapies

As noted in Section 3, conventional therapies for vitiligo have not been considered individually in the development of this review as they are beyond its scope. A summary of the conventional therapies considered in the expert consensus [39] and the emerging therapies reviewed is shown in Table 3.

## 6. Conclusions

Vitiligo is a disease with a complex and multifactorial pathogenesis, which has a great impact on the quality of life of patients.

New horizons are opening up in the treatment of this disease, both with long-known molecules such as 5-fluorouracil and with new molecules such as JAK inhibitors. The latter are postulated as a first-rate therapeutic tool for the treatment of vitiligo at present and in the near future, probably in conjunction with other traditional treatments such as UVB phototherapy.

At the moment, there is no clearly outstanding option or fully satisfactory treatment, so it is necessary to keep up the development of new drugs as well as the publication of long-term effectiveness and safety data for existing treatments.

## Figures and Tables

**Figure 1 ijms-24-17306-f001:**
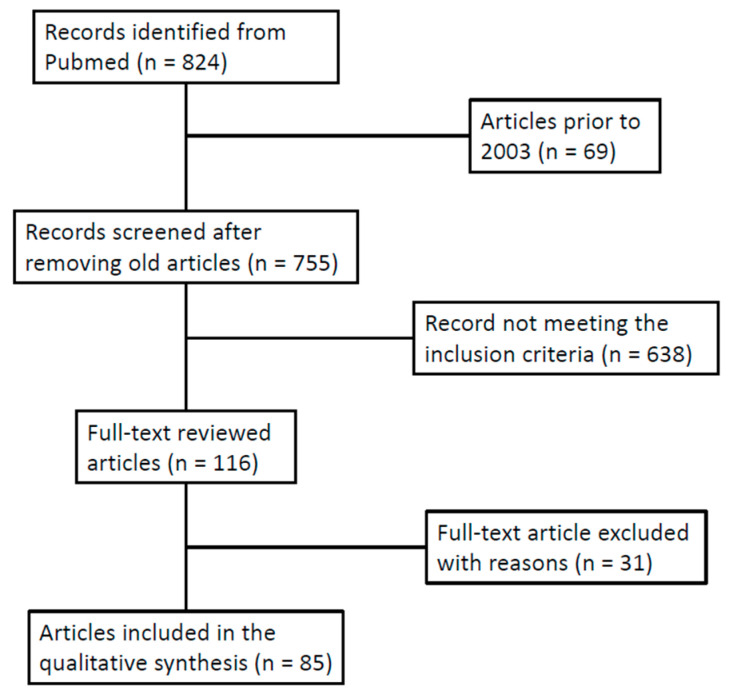
Flow chart showing the article selection process used in this narrative review.

**Table 1 ijms-24-17306-t001:** Case reports and case series reported to date on the use of iJAK for vitiligo.

Drug	Publication Data (Author/Year/Country)	Report Data (Drug and Route of Administration/Patients (n)/Treatment Duration/Area Affected/Follow-Up)	Outcome	Side Effects
Tofacitinib	Craiglow et al. [85]/2015/USA	5 mg tofacitinib citrate orally, initially 5 mg every other day; after 2 weeks, the dosage was increased to 5 mg/d 1 patient with forehead, torso and extremities vitiligo (10% BSA) treated and followed for 22 weeks (5 months)	Nearly complete repigmentation of the forehead and hands, partial repigmentation of extremities (5% BSA remained depigmented)	No adverse effects, no laboratory abnormalities
Ruxolitinib	Harris et al. [86]/2016/USA	20 mg ruxolitinib orally, twice daily. 1 patient with face, torso and extremities vitiligo treated for 20 weeks and followed for 16 more weeks	Improvement in facial pigmentation from 0.8% to 51%. 12 weeks after discontinuation of ruxolitinib, much of the regained pigment had regressed, from 51% to 16%	No side effects
Tofacitinib	Liu et al. [92]/2017/USA	5–10 mg tofacitinib orally QD-BID 10 patients treated for at least 3 months, an average of 9.9 months. 8 patients had generalized vitiligo and 2 patients had primarily acral involvement, with 1–100% BSA	A mean decrease of 5.4% BSA involvement with vitiligo was observed in 5/10 patients, while the other 5 patients did not achieve any repigmentation In the 5 patients who achieved some reversal of disease, repigmentation occurred only in sun-exposed areas of skin in 3 of them, diffusely in another patient undergoing concomitant full-body nbUVB phototherapy and to the dorsal hands in another patient after starting concomitant hand nbUVB phototherapy	Upper respiratory infection in 2 patients. 1 patient reported weight gain of 5 pounds and 1 patient reported arthralgias. Mild elevations of lipids were noted in 4 patients. There were no serious adverse events
Tofacitinib	Vu et al. [88]/2017/Australia	5 mg tofacitinib orally twice daily. 1 patient with multifocal vitiligo treated and followed for 6 months	Patient with concomitant atopic dermatitis and alopecia areata, both with great improvement. Marginal improvement in the vitiligo (decline in VASI score from 4.68 at baseline to 3.95 at 5 months)	Two episodes of self-resolving upper respiratory tract infections and diarrhea, no treatment interruption required
Ruxolitinib	Joshipura et al. [97]/2018/USA	Topical 1.5% ruxolitinib cream, twice daily. 2 patients with face, torso and extremities vitiligo treated for 38 and 12 weeks, respectively	Improvement in sun-exposed areas only (face and forearms)	No side effects
Tofacitinib	Gianfaldoni et al. [93]/2018/Italy	Tofacitinib citrate (10 mg orally every day) + cold light generator micro-focused phototherapy. 9 patients treated for at least 36 weeks	Repigmentation rate of 92% in the phototherapy + tofacitinib group, better than the phototherapy-alone group, in which only 72% obtained a repigmentation rate higher than 75%	No side effects
Tofacitinib	Kim et al. [89]/2018/USA	5 mg tofacitinib twice daily orally and narrow-band UV-B (360–500 mJ) 2 patients Patient 1: face (75% area affected), neck, torso and extremities vitiligo, results reported after 3 months Patient 2: face (90% area affected), torso and arms vitiligo, results reported after 6 months	Patient 1: complete repigmentation of her face, 75% or greater repigmentation of her neck, chest, forearms and shins, and only minimal freckling of the dorsal hands after full body phototherapy Patient 2: about 75% facial repigmentation. No repigmentation occurred at the other body sites (only facial phototherapy) Both had previously depigmented their faces using monobenzyl ether of hydroquinone (MBEH)	No side effects
Tofacitinib	McKesey et al. [98]/2019/USA	2% tofacitinib cream twice daily in conjunction with narrow-band ultraviolet B (NB-UVB) therapy thrice weekly 11 patients with face vitiligo treated for 12 ± 4 weeks and followed for a mean time of 112 days (range 84–154)	The mean facial VASI was 0.80 (range 0.1–2.25) at baseline and 0.23 (range 0.03–0.75) at follow-up, which is a mean improvement of 70% (range 50–87%)	No side effects
Tofacitinib	Mobasher et al. [99]/2020/USA	2% tofacitinib cream twice daily Concomitant treatment with topical steroids, topical calcineurin inhibitors, supplements (e.g., Polypodium leucotomos and Ginkgo biloba) or phototherapy was allowed 16 patients with “facial” or “non-facial” vitiligo followed for a mean time of 153 days (63–367)	13 experienced repigmentation with 4 patients experiencing > 90% repigmentation, 5 patients experiencing 25–75% repigmentation and 4 patients experiencing 5–15% repigmentation. 2 patients experienced no change and 1 patient experienced slow progression of depigmentation in the target lesion. Facial lesions improved more than non-facial lesions (*p* = 0.0216)	Acne-like papules on the face were reported by 1 patient. These lesions resolved with cessation of the medication. 1 patient reported subtle skin contour changes on his chin, which led to cessation of treatment after 2 weeks
Tofacitinib	Komnitski et al. [87]/2020/Brazil	5 mg tofacitinib orally twice daily. 1 patient with face, neck, elbows, hands and feet vitiligo, treated and followed for 104 weeks (2 years)	Complete repigmentation of the forehead and perilabial macules could be noted, as well as partial repigmentation in the posterior region of the neck and upper chest. No exposition to any source of ultraviolet radiation	No side effects
Tofacitinib	Olamiju et al. [100]/2020/USA	2% tofacitinib cream twice daily + narrow-band ultraviolet B phototherapy using a handheld unit 1 patient with face (segmental vitiligo) treated for 6 months and followed for 1 year	Freckling was observed within 4 weeks, almost complete repigmentation after 3 months and complete repigmentation at 6 months. The patient discontinued treatment after another month, and the area remained fully repigmented for approximately 6 months before a few depigmented macules began to reappear	No side effects
Ruxolitinib	Narla et al. [102]/2020/USA	1.5% ruxolitinib cream twice daily, for 2 patients presenting non-segmental vitiligo	Unspecified	Myalgias, which caused self-discontinuation in both patients. One of the patients presented mild elevation of the phosphokinase level (CPK)
Tofacitinib	Berbert-Ferreira et al. [101]/2021/Brazil	2% tofacitinib ointment twice daily only on facial lesions, combined with NB-UVB phototherapy, 3 times a week. The total dose for the face vitiligo was 1000 mJ/cm^2^ 1 patient presenting stable non-segmental vitiligo with acrofacial involvement treated for 9 months	Significant repigmentation of the forehead, nose, eyes and lips was observed	Minor adverse events such as erythema and transient acne
Tofacitinib	Fang et al. [94]/2021/Taiwan	5 mg tofacitinib orally once daily concomitant with nbUVB phototherapy. 4 patients with torso, arms, hands and leg vitiligo treated for 16 weeks	3 out of the 4 patients presented minimal or no change on vitiligo lesions. Only 1 of the patients had a partial response, with 14/42 (33%) of lesions showing signs of repigmentation. The results indicate that 5 mg daily tofacitinib concomitant with nbUVB phototherapy for 16 weeks is not sufficient for treating patients who showed an inadequate response to previous treatments	No side effects
Tofacitinib	Fang et al. [95]/2021/Taiwan	5 mg tofacitinib orally daily and 308 nm excimer light three times weekly 3 patients with torso, arms, hands, legs and feet vitiligo treated for 12 weeks	All patients had repigmentation and the mean reduction in VES was 32.7% (decreases of 38%, 44% and 16% in patients 1, 2 and 3, respectively). Of the 44 lesions, 14 (32%) showed follicular-patterned repigmentation, and of these, 6 repigmented lesions (43%) were in areas that were not sun-exposed regions. Acral lesions showed poor response	No side effects
Baricitinib	Li et al. [90]/2022/China	2 mg baricitinib orally twice daily + phototherapy + topical tacrolimus + topical steroids 2 patients with face, torso and extremities vitiligo treated for 6 months and 8 months, respectively	In patient 1, significant repigmentation after 8 months; in patient 2, over 75% repigmentation after 6 months	No side effects
Upadacitinib	Pan et al. [91]/2023/China	15 mg upadacitinib orally daily, combined with crisaborole. 1 patient with face, torso and extremities vitiligo treated for 4 months and followed for 7 months	After 4 months, there was nearly 90% repigmentation of his face and neck, 60% repigmentation of the chest and only a little repigmentation of the extremities	Worsening of acne

**Table 2 ijms-24-17306-t002:** A summary of the clinical trials reviewed for this section.

Drug	Study Data (Authors/Year/Country/NCT)	Study Design	Results	Side Effects
Ruxolitinib (topical)	Rothstein et al. [103]/2017/USA/NCT02809976	1.5% topical ruxolitinib cream, twice daily Single group, open-label, phase 2 11 patients followed for 20 weeks, presenting facial, upper limbs, torso or acral vitiligo	23% improvement in overall VASI in all patients. The 4 patients with facial involvement presented 76% improvement in fVASI. 3/8 patients responded on body surfaces. 1/8 responded on acral surfaces	Minor (erythema, hyperpigmentation and transient acne)
Ruxolitinib (topical)	Josahipura et al. [104] (open-label extension study of Rothstein’s)/2018/USA/No registration	1.5% topical ruxolitinib cream, twice daily (all 8 patients) + optional UVB phototherapy (3/8 patients chose it) Open-label extension study Phase 2 8 patients followed for 32 weeks, presenting facial, upper limbs, torso or acral vitiligo	Mean improvement in overall VASI of 37.6% ± 31.2% (*p* < 0.011). 5/8 had treatment response. 4 patients with facial vitiligo had mean 92% improvement. 3/6 had a response on their nonacral upper extremities (2 of these 3 had been treated with combination phototherapy). 2/3 patients (both of whom had opted for combination phototherapy) responded on the torso with a mean VASI improvement of 16.7% ± 16.7%	Minor (erythema and transient acne)
Ruxolitinib (topical)	Rosmarin et al. [84]/2020/USA/NCT03099304	Ruxolitinib cream (1.5% twice daily, 1.5% once daily, 0.5% once daily or 0.15% once daily) or vehicle (control group) twice daily A randomized, double-blind, dose-ranging study. Phase 2. 157 patients with vitiligo affecting at least 0.5% of the total body surface area (BSA) on the face and at least 3% of the total BSA on nonfacial areas; followed for 52 weeks	The primary endpoint at week 24, F-VASI50, was reached by significantly more patients given the two highest doses of ruxolitinib cream (1.5% twice daily, 15 (45%) of 33 patients, odds ratio (OR) 24·7, 95% CI 3.3–1121.4; *p* = 0.0001; 1.5% once daily, 15 (50%) of 30 patients, OR 28.5, 95% CI 3.7–1305.2; *p* < 0.0001) and also by more patients who received the two lowest doses of ruxolitinib cream (0.5% once daily, eight (26%) of 31; 0.15% once daily, ten (32%) of 31) compared with the vehicle (one (3%) of 32 patients). T-VASI50 at week 52, a key secondary endpoint, was reached by patients in the total population in a dose-dependent manner (1.5% twice daily, 12 (36%) of 33; 1.5% once daily, 9 (30%) of 30; 0.5% once daily, 8 (26%) of 31)	Application site pruritus was the most common treatment-related adverse event among patients given ruxolitinib cream (1 (3%) of 33 in the 1.5% twice daily group; 3 (10%) of 30 in the 1.5% once daily group; 3 (10%) of 31 in the 0.5% once daily group; and 6 (19%) of 31 in the 0.15% once daily group), with 3 (9%) of 32 patients showing application site pruritis in the control group. Acne was noted as a treatment-related adverse event in 13 (10%) of 125 patients who received ruxolitinib cream and 1 (3%) of 32 patients who received vehicle cream. All treatment-related adverse events were mild or moderate in severity and similar across treatment groups. No serious adverse events were related to study treatment
Ruxolitinib (topical)	Hamzavi et al. [105]/2022/USA/NCT03099304	Ruxolitinib cream (1.5% twice daily, 1.5% once daily, 0.5% once daily or 0.15% once daily) or vehicle (control group) twice daily A randomized, double-blind, dose-ranging study. Phase 2. 157 patients with vitiligo affecting at least 0.5% of the total body surface area (BSA) on the face and at least 3% of the total BSA on nonfacial areas; followed for 52 weeks	Among patients with vitiligo affecting ≤ 20% of T-BSA at baseline, both doses of ruxolitinib cream (1.5% once daily and twice daily) produced notable T-VASI50 and T-VASI75 responses at week 52. The 1.5% ruxolitinib cream twice-daily dose produced the highest proportion of T-VASI50 responders in the head/neck region (60.0%), followed by the upper and lower extremities (52.9% and 52.6%, respectively). T-VASI50 of the hands and feet was noted for 15.0% and 29.4% of patients, respectively, who received 1.5% ruxolitinib cream twice daily	Unspecified
Ruxolitinib	Pandya et al. [106]/2022/USA/NCT03099304	Ruxolitinib cream with concomitant narrow-band UVB (NB-UVB) phototherapy during the open-label phase after week 52 A randomized, double-blind, dose-ranging study. Phase 2. 19 patients with vitiligo affecting at least 0.5% of the total body surface area (BSA) on the face and at least 3% of the total BSA on nonfacial areas; followed for 52 weeks	After the addition of NB-UVB phototherapy, F-VASI and T-VASI scores improved in 15 of 19 patients (78.9%) and 18 of 19 patients (94.7%), respectively. In these 19 patients, the mean percentage improvement at week 104 was 50.2% for F-VASI and 29.5% for T-VASI versus the improvement at the last visit before the addition of NB-UVB phototherapy. Postcombination therapy response parameters were similar to data at week 104 from 70 patients who remained on ruxolitinib cream alone from day 1; responses were higher at week 104 versus week 52 among patients who received ruxolitinib cream alone	No adverse events considered to be related to the treatment
Ruxolitinib	Rosmarin et al. [107]/2022/USA/TRuE-V1 (NCT04052425) and TRuE-V2 (NCT04057573)	1.5% ruxolitinib cream or matching vehicle cream twice daily to all depigmented vitiligo lesions, on a 2:1 ratio Two multinational, phase 3, double-blind, vehicle-controlled trials of identical design conducted across 101 centers 661 patients (330 TRuE-V1 and 331 TRuE-V2) with face and body vitiligo followed for 24 weeks	In TRuE-V1, the percentage of patients with an F-VASI75 response at week 24 was 29.8% in the ruxolitinib cream group and 7.4% in the vehicle group (relative risk, 4.0; 95% confidence interval (CI), 1.9 to 8.4; *p* < 0.001). In TRuE-V2, the percentages were 30.9% and 11.4%, respectively (relative risk, 2.7; 95% CI, 1.5 to 4.9; *p* < 0.001). The results for key secondary end points showed superiority of ruxolitinib cream over vehicle control (F-VASI50, F-VASI90, T-VASI50 and F-VASI75 in the 52-week extension study)	Among patients who applied ruxolitinib cream for 52 weeks, adverse events occurred in 54.8% in TRuE-V1 and 62.3% in TRuE-V2; the most common adverse events were application site acne (6.3% and 6.6%, respectively), nasopharyngitis (5.4% and 6.1%) and application site pruritus (5.4% and 5.3%)
Tofacitinib (oral)	Song et al. [111]/2022/China/No registration	Oral tofacitinib at 5 mg twice daily. Both control and treatment group were treated with halometasone cream applied externally to the lesions on the torso and limbs twice a day, and 0.1% tacrolimus ointment or pimecrolimus cream applied externally on the face and neck twice a day. In addition, NB-UVB therapy was administered three times weekly for a period of 16 weeks Real-world clinical practice out-of-label tofacitinib clinical trial 15 patients in treatment group and 19 controls with face and body vitiligo followed for 16 weeks	From eighth week, the repigmentation level was significantly higher in the combination than the control group (*p* < 0.05). The repigmentation improved in the tofacitinib group on acral lesions, torso and extremities No significant differences in lesions on the face and neck were observed between the combination and control groups during 16 weeks of treatment (*p* > 0.05), probably because both groups had great improvement	One patient treated with tofacitinib developed mild pain in his right thumb and right hallux after 3 weeks of treatment, but the pain resolved with cessation of tofacitinib 1 week later. Mild effects related to phototherapy
Ritlecitinib (oral)	Ezzedine et al. [110]/2023/USA/NCT03715829	Patients were randomized to once-daily oral ritlecitinib ± 4-week loading dose (200/50 mg, 100/50 mg, 30 mg or 10 mg) or placebo for 24 weeks (dose-ranging period). 187 patients subsequently received ritlecitinib at 200/50 mg daily in a 24-week extension period Phase 2b, randomized, double-blind, placebo-controlled, parallel-group, multicenter and dose-ranging study 364 patients with face and body vitiligo treated for a 24-week dose-ranging period and 24-week extension period	Significant differences from placebo in percent change from baseline in Facial-Vitiligo Area Scoring Index were observed for the 50 mg ritlecitinib groups with (−21.2 vs. 2.1; *p* < 0.001) or without (−18.5 vs. 2.1; *p* < 0.001) a loading dose and 30 mg ritlecitinib group (−14.6 vs. 2.1; *p* = 0.01). Accelerated improvement was observed after treatment with 200/50 mg ritlecitinib in the extension period (n = 187)	The 3 most common TEAEs were nasopharyngitis (15.9%), upper respiratory tract infection (11.5%) and headache (8.8%). 4 patients had confirmed cases of herpes zoster (all non-serious), 2 patients had malignancies (nonmelanoma skin cancers) and there were no thromboembolic events. No serious adverse events
Upadacitinib (oral)	No results published/2023/USA/NCT04927975 [112]	Oral upadacitinib (dose ranging) vs. placebo A multicenter, randomized, double-blind, placebo-controlled, dose-ranging study to evaluate the safety and efficacy of upadacitinib in subjects with non-segmental vitiligo. Phase 2 185 patients with face and body vitiligo followed for at least 24 weeks, up to 52 weeks	Ongoing. No results published	Ongoing. No results published
Ruxolitinib (topical)	No results published/2023/USA/NCT05247489 [109]	Group A: 1.5% ruxolitinib cream + narrow-band ultraviolet B phototherapy (NB-UVB). Group B: 1.5% ruxolitinib cream monotherapy A randomized, phase 2, open-label interventional study. 55 patients with face and body vitiligo follows for 48 weeks	Ongoing. No results published	Ongoing. No results published
Ruxolitinib (topical)	No publications available/2023/USA/NCT04530344 [108]	1.5% ruxolitinib cream or matching vehicle cream twice daily A double-blind, vehicle-controlled, randomized withdrawal and treatment extension study to assess the long-term efficacy and safety of ruxolitinib cream in participants with vitiligo Phase 3 458 patients with face and body vitiligo followed for 52 weeks	Completed. No publications available	Completed. No publications available
ARQ-252 (topical)	No results published/2022/USA/NCT04811131 [113]	0.3% ARQ-252 cream BID or vehicle cream BID, and active phototherapy or sham phototherapy for 24 weeks Phase 2a, parallel-group, double-blind, vehicle-controlled study of the safety and efficacy of 0.3% ARQ-252 cream in combination with NB-UVB phototherapy treatment in subjects with non-segmental facial vitiligo 114 patients with face and body vitiligo followed for 24 weeks	Terminated. No publications available	Terminated. No publications available
Cerdulatinib (topical)	No publications available/2022/USA/NCT04103060 [114]	0.37% cerdulatinib gel applied topically twice daily vs. vehicle cream A phase 2a, randomized, double-blind, vehicle-controlled study to assess the safety, tolerability and systemic exposure of 0.37% cerdulatinib gel in adults with vitiligo 33 patients with face and body vitiligo followed for 6 weeks	No publications available	No publications available
ATI-50002 (topical)	No publications available/2020/USA/NCT03468855 [115]	ATI-50002 topical solution, high dose active, twice daily, 24 weeks An open-label pilot study of the safety, tolerability and efficacy of ATI-50002 topical solution administered twice daily in adult subjects with non-segmental facial vitiligo. Phase 2 34 patients with face vitiligo followed for 24 weeks	Mean change in facial depigmentation in quantified area of interest (AOI) from baseline (Visit 2) to week 24 worsened after treatment: mean change + 2 (standard deviation 8.41)	Alcoholic pancreatitis and acute myocardial infarction (in 1 patient, not related to the drug), application site acne, other minor local adverse events

**Table 3 ijms-24-17306-t003:** Conventional [39] versus emerging therapies.

Treatment Modality	Conventional versus Emerging	Advantages/Data That Favor Its Use	Main Disadvantages
Topical corticosteroids (TCSs)	Conventional	Recommended for vitiligo, particularly for extrafacial locations and more limited treatment areas Wide experience on its use	More effective for stabilization of vitiligo than for repigmentation Local side effects if applied continuously (skin atrophy, telangiectasia, hypertrichosis, acneiform eruptions and striae)
Topical calcineurin inhibitors	Conventional	As affective as TCS on face and neck, with better safety profile in these locations No serious adverse events detected in patients with vitiligo treated with topical calcineurin inhibitors	Less effective than TCS on extrafacial lesions Off-label use
Narrow-band ultraviolet B phototherapy (NB-UVB)	Conventional	Preferred first-line therapy for widespread or rapidly progressing disease No significant association with greater incidence of basal cell carcinoma, squamous cell carcinoma or melanoma	Bad response of acral lesions and areas lacking melanocyte reservoir Erythema and xerosis are common Multiple sessions are required, so patients have to attend their healthcare center two or three times a week for several months
Excimer devices	Conventional	Equally effective or even superior compared to NB-UVB Safety and tolerability of excimer laser therapy is comparable to NB-UVB	The cost of therapy is higher than NB-UVB Long-term adverse events not well-established
Home phototherapy	Conventional	Better compliance, similar repigmentation outcomes, similar frequency of adverse effects and less time investment	Shortage of home phototherapy units, high initial cost, low energy output of the device over time, lack of mechanical servicing and unfamiliarity of patients with the modality
Oral steroid minipulse therapy (dexamethasone, metilprednisolone or prednisone)	Conventional	Useful to stop disease progression	Not suitable for repigmentation on monotherapy Relapse after discontinuation Systemic corticosteroid-class side effects: weight gain, insomnia, agitation, acne, menstrual disturbances, hypertrichosis, growth retardation in children and immunosuppression
Surgical interventions	Conventional/emerging	Many different techniques A treatment option for segmental vitiligo and other localized and stabilized forms of vitiligo (non-segmental) after the documented failure of medical interventions	Koebner phenomenon is possible High cost Pros and cons depend on the technique, but this topic exceeds the subject of this review and should be discussed separately
Afamelanotide	Conventional	Potential benefit for use in combination with phototherapy	Subcutaneous administration More data need to be collected
Cyclosporine	Conventional	Useful for arresting vitiligo progression Might be useful as adjunctive treatment in autologous noncultured melanocyte–keratinocyte cell transplantation procedure	Not suitable for long-term treatment
Phosphodiesterase 4 (PDE-4) inhibitors	Emerging	Case reports of improvement with apremilast or crisaborole in monotherapy	Conflicting data on its use in combination with phototherapy
Trichloroacetic acid	Emerging	Good response on face vitiligo in combination with microneedling or phototherapy	More data need to be collected
Basic fibroblast growth factor (bFGF)	Emerging	Could improve repigmentation when combined with phototherapy or tacrolimus ointment	More data need to be collected
TNF inhibitors	Emerging	Isolated case reports showing efficacy in repigmentation	Most studies show no response or even TNFα inhibitor-induced vitiligo
Secukinumab	Emerging	A suitable option to replace a TNFα inhibitor after new-onset vitiligo related to TNFα inhibitors	Not recommended for the treatment of isolated vitiligo
Pseudocatalase	Emerging	Oxidative stress plays a role in vitiligo pathogenesis Initial data supported its efficacy	All recent data show no improvement in repigmentation
JAK inhibitors	Emerging	Multiple case reports and clinical trials support its efficacy Ruxolitinib cream already approved for vitiligo Several ongoing clinical trials with promising results	Long-term efficacy and long-term safety data need to be assessed More expensive than conventional treatments Results when used in combination with other treatment modalities need to be studied
5-fluorouracil	Emerging	Useful to achieve repigmentation when used alongside phototherapy, microneedling and dermabrasion Intradermal infiltrations of 5-FU have also been tested	Local side effects (burning, pruritus, blistering)
Platelet-rich plasma	Emerging	Synergistic effect in conjunction with laser, phototherapy and surgical treatments	Limited data on monotherapy More data need to be collected
Microneedling	Emerging	Could improve repigmentation in monotherapy or when combined with phototherapy or 5-fluorouracil	More data need to be collected

## Data Availability

The data that support the findings of this study are available from the corresponding author upon reasonable request. The data are not publicly available due to privacy or ethical restrictions.

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
