# Peer review of "Vitiligo: Pathogenesis and New and Emerging Treatments"

_ijms, 2023, doi:10.3390/ijms242417306_

Round 1
Reviewer 1 Report
Comments and Suggestions for Authors
The review by Perez-Bootello is interesting, however several aspects of this manuscript could be improved especially with regard to the description of pathogenic mechanisms.
My suggestions:
-the treatment of oxidative stress is treated completely separately from the autoimmune aspect while it is believed that intrinsic and extrinsic mechanisms of oxidative stress can function as a trigger for an inflammatory/autoimmune reaction.
-consideration of numerous evidences of metabolic imbalances in pathogenic mechanisms is totally lacking, again metabolic impairment are considered important in generating second messengers implicated in local inflammation.
-as far as therapeutic approaches are concerned, it seems to me that excessive emphasis is given to the use of pseudocatalase, which has not found significant clinical use over the years. Most of the studies are not recent.
The authors should also consider emerging possibilities for using regenerative therapies for vitiligo such as microneedling, PRP, cell transplantation…….
Tables need to be shortened
Reviewer 2 Report
Comments and Suggestions for Authors
The article is a narrative review of the state of the art of currently available vitiligo therapies, excluding physical therapies outside of phototherapy and surgical therapies.
The idea is interesting, and the work is well written, still there are many works in the literature that summarize emerging or actual treatments for vitiligo.
I would ask, after reading the introductive section on conventional why only phototherapy was specified. Therefore, the other known conventional therapies, especially topical ones, would be missing (Steroid, topical tacrolimus, etc.) would be missing. Is there a way to introduce them?
Is there a reason, then, why some conventional treatment have been or will be described and chosen (a search was carried out on the Pubmed database with an ad hoc string ? Have some guidelines been taken as reference?) I think would be useful for the reader in order to better articulate and justify the initial part.
Furthermore to collect the articles on the emerging treatments a search string with the names of the remedies that will be then described have been used. This happened without introducing how these drugs have been chosen in the first instance.
It would be useful to better clarify how some emerging treatments have been chosed rather than others.
Reviewer 3 Report
Comments and Suggestions for Authors
Title: interesting, comprehensive and concise
Abstract line 14: Maybe the authors mean "analysis" and not "synthesis"?
Line 18: I suggest to start the sentence: "based on the qualitative analysis, it can be concluded that there is no clear superiority of the any of the reviews treatments…….."
Introduction: some part are missing: other review in the field, the objective of this review and what is its contribution compared to other studies?
Pat attention to the subtitle, sometimes they are numerated and sometimes not.
Section 2.1. . Autoinmunity
Then the following subsection is not numerated.
The following style of subtitles is not recommended "Vitiligo is a disease primarily driven by cell-mediated immunity" instead the authors can write the following" Cell mediated driven vitiligo" and so on for the following subtitles
The following subtitle doesn't belong to the "Autoinmunity"
Subtitles are usually not ended by a full stop.
3. Literature search: this section looks good and well presented.
Add a table (shot one) to summarize the current treatments: conventional vs. emerging therapies with a special focus on the advantages and disadvantages of each treatment
Tables 1 and 2 are very hard to read some columns can be deleted and the data can be merged with other columns, for example in Table 2: Route of administration, Treatment, Allocation and phase, Status, Duration (week s), n and Area can all be under one column entitled " study design" the same for table 1.
Are there any trials on combination therapy?? If yes, they should be presented in a separate section with a summarizing table.
Comments on the Quality of English Language
Minor editing of English language required
Round 2
Reviewer 1 Report
Comments and Suggestions for Authors
The present form of the manuscript is overall improved and I have not additional comments
Reviewer 3 Report
Comments and Suggestions for Authors
Accepted in the present form